# Femoral Structure and Biomechanical Characteristics in Sanfilippo Syndrome Type-B Mice

**DOI:** 10.3390/ijms241813988

**Published:** 2023-09-12

**Authors:** Frederick James Ashby, Evelyn J. Castillo, Yan Ludwig, Natalia K. Andraka, Cong Chen, Julia C. Jamieson, Nadia Kabbej, John D. Sommerville, Jose I. Aguirre, Coy D. Heldermon

**Affiliations:** 1Department of Medicine, University of Florida, Gainesville, FL 32611, USA; yludwig@ufl.edu (Y.L.); natalia.andraka@medicine.ufl.edu (N.K.A.); juliajamieson@ufl.edu (J.C.J.); nadia.kabbej@medicine.ufl.edu (N.K.); johnsommerville@ufl.edu (J.D.S.); coy.heldermon@medicine.ufl.edu (C.D.H.); 2Department of Physiological Sciences, University of Florida, Gainesville, FL 32611, USA; evelynjcastillo@ufl.edu (E.J.C.); aguirrej@ufl.edu (J.I.A.); 3Department of Orthopaedic Surgery & Sports Medicine, University of Florida, Gainesville, FL 32611, USA; chenc@ortho.ufl.edu

**Keywords:** Sanfilippo syndrome, MPS IIIB, lysosomal storage disease, bone, model

## Abstract

Sanfilippo syndrome Type-B, also known as mucopolysaccharidosis IIIB (MPS IIIB), accounts for approximately one-third of all Sanfilippo syndrome patients and is characterized by a similar natural history as Type-A. Patients suffer from developmental regression, bone malformation, organomegaly, GI distress, and profound neurological deficits. Despite human trials of enzyme replacement therapy (ERT) (SBC-103, AX250) in MPS IIIB, there is currently no FDA approved treatment and a few palliative options. The major concerns of ERT and gene therapy for the treatment of bone malformation are the inadequate biodistribution of the missing enzyme, N-acetyl-α-glucosaminidase (NAGLU), and that the skeleton is a poorly hit target tissue in ERT and gene therapy. Each of the four known human types of MPS III (A, B, C, and D) is usually regarded as having mild bone manifestations, yet it remains poorly characterized. This study aimed to determine bone mineral content (BMC), volumetric bone mineral density (vBMD), and biomechanical properties in femurs MPS IIIB C57BL/6 mice compared to phenotypic control C57BL/6 mice. Significant differences were observed in MPS IIIB mice within various cortical and cancellous bone parameters for both males and females (*p* < 0.05). Here, we establish some osteogenic manifestations of MPS IIIB within the mouse model by radiographic and biomechanical tests, which are also differentially affected by age and sex. This suggests that some skeletal features of the MPS IIIB mouse model may be used as biomarkers of peripheral disease correction for preclinical treatment of MPS IIIB.

## 1. Introduction

Mucopolysaccharidosis (MPS) is a cluster of lysosomal storage diseases that affect the breakdown of glycosaminoglycans (GAGs), leading to each MPS type’s molecular and clinical sequelae [1]. MPS diseases are categorized as Type-I, Type-II, Type-III, Type-IV, Type-VI, and Type-VII based on which enzyme in the respective GAG pathway is affected [2]. Type-V (Scheie syndrome) was later found to be a milder form of Type-I (Hurler syndrome), which affects the *IDUA* gene pathway. MPS I is known as Hurler’s syndrome, and MPS II, known as Hunter’s syndrome, are characterized by the accumulation of dermatan sulfate and heparan sulfate [3]. MPS Type-III, also known as Sanfilippo syndrome, is characterized by an inability to breakdown heparan sulfate alone. MPS VII, also known as Morquio syndrome, is characterized by the accumulation of dermatan sulfate, heparan sulfate, and chondroitin sulfate. It is worth noting that MPS Type-I, II, III, and VII all have heparan sulfate as a common GAG accumulation [2].

All MPS types are afflicted with skeletal malformations, yet much remains to be elucidated about the mechanism of this. Heparan sulfate and other GAGs have been shown to stimulate Toll-like Receptor 4 (TLR-4) in human dendritic cells [4,5], resulting in the production of many cellular cytokines, such as TNF-α, that mediate neuroinflammation and abnormal bone ossification due to enhanced chondrocyte turnover [6,7,8,9]. In MPS I, II, and VI, a high TNF-α was shown to be associated with increased disability and pain [10]. It is also speculated that vitamin D deficiency plays a key role in osteological manifestations of MPS [11]. Despite ERT and stem cell transplant being available for some MPS subtypes, currently, there are no accepted treatments for the bone disease manifestations of MPS [12,13,14]. Orthopedic complications of MPS III in particular remain an area poorly characterized, in part because of the rarity of the disease and the perceived mildness of osteological defects compared to other MPS types [15,16]. Additionally, the effects of gene therapy and stem cell transplant seem only helpful for neurologic disease manifestations when incorporated before age 2. MPS IIIB mice also demonstrate increased TNF-α production [17] with severe early inflammatory effects in the brain. TLR4-knockout MPS IIIB mice display early attenuation of neuroinflammation. However, the storage component still progresses and ultimately results in cell death with the activation of alternative inflammatory pathways later in life [17].

In MPS III, only a small sample of clinical observational studies with MPS III have shown impaired growth [18,19]. Clinical osteological manifestations of MPS III in humans involve mostly hip and spine dysplasia [11,20], with common complications including scoliosis and femoral head osteonecrosis [21]. The MPS IIIB C57BL/6 mouse model develops blunting of the face and thickened phalanges, yet the long bone effects have yet to be evaluated. It has previously been shown that MPS VII mice have increased bone mineral density (BMD) and bone mineral content (BMC) [22]. The cause of bone pathology in MPS IIIB and other MPS III subtypes is an area yet to be well-characterized. We hypothesized that the MPS IIIB mouse would have quantifiable changes in femur radiographic biometrics, such as BMD and BMC, and that these would lead to differences in the biomechanical properties of the femurs. 

## 2. Results

### 2.1. Peripheral Quantitative Computed Tomography (pQCT) Analysis of Femurs

#### 2.1.1. Mid-Diaphyseal qQCT

Femurs of 3-month-old mice were scanned at mid-diaphysis (approximately 50% of total femur length), analyzed using Stratec software macros previously described [22], and compared between MPS IIIB mice and sex-matched phenotypic controls (Figure 1) via two-way ANOVA considering sex and genotype and the interaction between the two. Post hoc analysis was performed with a Dunn–Šidák correction. Before epiphyseal closure, MPS IIIB mice demonstrated a total mid-diaphyseal BMC that was increased by 15.9% in males (*p* < 0.0001) and 11.1% in females (*p* < 0.001). However, there was no observable difference in BMD for either sex. Mid-diaphyseal cortical thickness for MPS IIIB was increased by 8.2% in males. However, no statistically significant difference was observed in females. The total cortical area was increased in MPS IIIB mice by 14.1% in males (*p* < 0.0001) and 9.7% in females (*p* < 0.001), and this was related to a 6.0% increase in both sexes for periosteal circumference (*p* < 0.001). Endosteal circumference in MPS IIIB mice for males increased by 4.6%, but this did not reach statistical significance and increased by 8.7% in females (*p* < 0.05). 

The effect of MPS IIIB on mid-diaphyseal biophysical properties by pQCT after epiphyseal closure was investigated in 6-month-old and 12-month-old mice to assess the longitudinal effects on bone characteristics. The data from 3-month-old mice were included for comparison, and all data were analyzed by two-way ANOVA considering age and genotype and the interaction between the two. Post hoc analysis was performed with a Dunn–Šidák correction. Due to the considerable difference between male and female bone morphology, each gender was analyzed separately to maintain comparability. 

Mid-diaphyseal BMC and BMD demonstrated profound changes in MPS IIIB mice for both sexes (Figure 2). Total mid-diaphyseal BMC in MPS IIIB mice was increased in males and females throughout life compared to their respective controls (*p* < 0.05). However, this effect became less pronounced in males after 12 months of age and only became more pronounced in females (*p* < 0.0001). When considering mid-diaphyseal BMD in MPS IIIB mice, however, both sexes were indistinguishable from their phenotypic controls at 3 months and were consistently elevated (*p* < 0.05) at 6 months and 12 months. This increased BMD with an MPS IIIB genotype was more pronounced in female mice (approximately 13% increase; *p* < 0.001) than male mice (around 4% increase; *p* < 0.05). Mid-diaphyseal morphology (Figure 3) demonstrated changes in MPS IIIB mice in both sexes respective to their controls. Male MPS IIIB mice had greater bone cortical area than control male mice (*p* < 0.001), yet this difference disappeared with age, particularly after 12 months (ns differences). In contrast, the female MPS IIIB mice had greater cortical area than control mice and became more pronounced over time (*p* < 0.001). Mid-diaphyseal cortical thickness in the male MPS IIIB mice was slightly increased compared to their controls (*p* < 0.05), yet in females, this effect again became more profound over time (*p* < 0.001).

#### 2.1.2. Metaphyseal pQCT

Femurs of 3-month-old mice were scanned at the metaphyseal (approximately 25% of the total femur length) and compared between MPS IIIB mice and sex-matched phenotypic controls (Figure 4). MPS IIIB mice had greater total BMC than the controls, with a 47.9% increase in males (*p* < 0.0001) and 16.8% in females (*p* < 0.01), respectively. Metaphyseal total BMD was also greater in MPS IIIB mice than in the controls, with a 19.8% increase in males (*p* < 0.0001) and a 6.9% in females, respectively. However, the female cohort did not reach statistical significance. This was further investigated to assess the effects on the metaphyseal cancellous bone compartment. We found that BMC was 110.7% greater (*p* < 0.0001) in males and 47.9% greater in male and female MPS IIIB mice than their respective controls. However, the female cohort did not reach statistical significance. Trabecular BMD was 71.6% greater in male MPS IIIB mice (*p* < 0.0001) and 38.5% greater in female MPS IIIB mice (*p* < 0.01) than in their respective control groups. Both metaphyseal total area and trabecular area were increased by about 23% in male MPS IIIB mice (*p* < 0.0001). Still, no statistically significant increase was observed in female MPS IIIB mice (ns).

The effect of MPS IIIB on metaphyseal biophysical properties after epiphyseal closure was investigated in 6-month-old and 12-month-old mice to assess longitudinal effects on bone characteristics. We found profound changes in metaphyseal BMC and BMD in MPS IIIB mice of both genders (Figure 5). Total metaphyseal BMC was greater in male and female MPS IIIB mice throughout life compared to their respective controls (*p* < 0.01). However, this effect also became less pronounced in males after 12 months of age and became pronounced in females (*p* < 0.0001). Furthermore, we found that the femurs of male MPS IIIB mice had greater metaphyseal BMD than the control mice, and this effect became less pronounced over time.

Metaphyseal trabecular BMD and BMC (Figure 6) demonstrated profound changes in MPS IIIB mice in both sexes respective to their controls throughout life. Again, we see an increase in the difference between MPS IIIB mice compared to their controls after the closure of the secondary growth plate, while in males, the difference to their controls becomes less pronounced.

### 2.2. Biomechanical Analysis of Femurs

Analyses from the 3-point bend test demonstrated distinct differences in MPS IIIB femurs (Figure 7). In male MPS IIIB mice, the ultimate load (maximum force) was increased early in life. Still, this difference disappeared after six months of age (Figure 7a). In females, the top load was not raised during early life expansion but later in life (Figure 7a). Work to fracture was notably increased in the male MPS IIIB mice throughout life. However, no difference was seen in the female MPS IIIB mice (Figure 7b). Ultimate stress showed no difference in the MPS IIIB mice of either sex (Figure 7c) and stiffness only showed a difference in the 12-month-old female MPS IIIB mice (Figure 7d). When the ultimate load was correlated to respective radiographic values (Figure 8a–c), there was a moderate correlation between ultimate load and cortical area, cortical thickness, and cortical BMD (R^2^ = 0.48–0.51; *p* < 1 × 10^−12^). When stiffness was correlated to these same radiographic values (Figure 8d–f), there was a statistically significant weak correlation (R^2^ = 0.19–0.27; *p* < 1 × 10^−4^).

## 3. Discussion

### 3.1. Summary

The results ultimately demonstrated detectable physical changes in the femurs of MPS IIIB mice compared to age- and sex-matched controls. These changes in MPS IIIB mice were influenced by both the mouse’s sex and age. Male MPS IIIB mice demonstrated an increased femoral BMD, which was more pronounced in the metaphyseal trabecular bone, and increased diaphyseal cross-sectional compared to their peers. This effect generally diminished as the mice aged. Female MPS IIIB mice, interestingly, had a similar difference compared to their peers that generally became more profound as the mice aged. This was similar to what has previously been reported in MPS VII mouse femurs [22]. Interestingly, both disease mouse models are able to feed themselves unabated, suggesting that orthopedic manifestations of these diseases may be partially caused by mechanisms other than nutritional deficiency, which is believed to be the cause of MPS III osteopenia in humans. Overall, these radiological changes in the mice correlated with the biomechanical properties of the femurs, particularly ultimate load. Interestingly, male MPS IIIB mice showed a statistically significant increase in work to fracture throughout all ages. Anecdotally, it was observed that male MPS IIIB mouse femurs generally took longer to fracture completely compared to other groups, contributing to this observed increase. Thus, the time point from the ultimate load to the point of fracture was generally longer in this group. While this study was not designed to investigate the mechanism for these unexpected sex differences, it is speculated that sexual dimorphic differences in the osteological phenotype of MPS IIIB mice could be related to the estrogen modulation of osteoclasts in bone turnover [23]. 

When taken together, we demonstrated that the MPS IIIB C57BL/6 mouse has measurable femur radiologic and biomechanical changes in both male and female mice; however, the differences can vary greatly depending on age and biological sex. This underscores the critical importance of basic experimental principles in animal research: ensuring proper sex stratification in MPS IIIB mouse studies and ensuring similar ages among the experimental groups being compared. It is not known if these dimorphic phenotypes between males and females are analogous to dimorphisms in humans with MPS IIIB. Regardless, these data support our hypothesis that there are measurable changes in the femurs of MPS IIIB C57BL/6 mice compared to their respective controls. 

A key limitation of our study is that we were unable to evaluate the relative physical activity of the mice throughout life. This may plausibly contribute to the sex differences observed here. We also were not able to evaluate non-long bones, which is where a considerable portion of the clinical manifestation of MPS IIIB in humans is observed. Lastly, some age groups had a higher sample size than others due to premature animal death in some groups, as well as differences in the availability of animals in each age/sex category, which was not ideal. However, it should be noted that all pQCT groups were sufficiently powered based on an initial pilot (α = 0.05, β = 0.20).

### 3.2. Future Work

Previous animal research has suggested that certain clinical manifestations of MPS IIIB may be partially caused by overactivation of the TLR-4 cascade [17], largely driven by the PAMP/DAMP recognition of GAGs, such as heparan sulfate. Other studies have suggested that this pathophysiology may be relevant to different MPS types [6,7,8,9], with clinical studies implicating TNF-α as a prognostic value in MPS [10] and perhaps even a therapeutic target [24,25]. It is worth noting that a study of TLR4-KO-MPS-IIIB mice showed cytokine levels similar to wild type until after three months of age [9]; however, these mice eventually developed abnormal oxidative stress, putatively leading to neurodegeneration independent of innate inflammatory pathways. 

A delay in the inflammatory contribution to MPS IIIB neurodegeneration may allow improved development of neural pathways during early development. It is possible that peripheral manifestations of MPS IIIB, such as abnormal bone development, may also be ameliorated by reducing inflammation, thus potentially delaying disability and reducing pain. However, at this time, the role of the TLR4 pathway in bone formation in MPS III is currently unknown [26]. Furthermore, for translational research in MPS IIIB, we found that it is not well-established how to assess if osteogenic manifestations of the disease are being corrected or if they even exist in an MPS IIIB mouse model. Only a small sample of clinical observational studies with MPS III have shown impaired growth [18,19]. Currently, there is no standard of care for treating the bone manifestations of MPS [12]. In this study, we present evidence that the MPS IIIB mouse model possesses consistent radiological disparities compared to its phenotypically normal counterpart and demonstrates biomechanical differences as a result. These biomechanical properties had a moderate to weak correlation with the radiological metrics. Overall, we consider the MPS IIIB C57BL/6 mouse used here to be a potential model for the osteological manifestations of MPS IIIB, as evident by the differences in radiological and biomechanical properties compared to the control mice. 

A possible intervention to address bone pathology and simultaneously test the hypothesis that bone manifestations of MPS IIIB are partially driven by TLR4 overactivation would be a simple small-molecule TLR4 or TLR pathway inhibitor. Resatorvid is a TLR-4 antagonist [27,28,29] that crosses the blood–brain barrier (BBB) and has shown activity in hypoxic and traumatic brain injury models [30]. Additionally, Resatorvid (TAK-242) has demonstrated disease modification in mouse models of arthritis, indicating our ability to affect bone and joint physiology [31]. This drug has already been through phase-I/II clinical trials in humans [32]. It would potentially be a quick solution for FDA approved palliative care for patients with Sanfilippo syndrome. The original trials targeted sepsis patients and were unsuccessful in reaching their clinical endpoints but demonstrated relative safety. A TLR4 antagonist with a more favorable side effect profile would be ApTOLL [33], a TLR-4 targeting DNA aptamer. Unfortunately, ApTOLL is not commercially available and is patented (WO2020/230108), so it could not be incorporated into a preclinical study as easily. Alternatively, TLR pathway inhibitors such as dasatinib, emavusertib, ruxolitinib, or pacritinib, which currently are approved or in the clinical phase for various cancers, may provide a broader reduction in inflammation and a more rapid clinical translation. The risk of this approach is a more extensive immune suppression that may increase susceptibility to infection. Dasatinib acts downstream of TLR4, TLR3, and TLR9 signaling [34]. It has a known safety profile in pediatric patients and could be repurposed easily. Pacritinib is approved for myelofibrosis and inhibits IRAK1 and downstream activation from TLR4, though pediatric studies have not yet been completed. Ruxolitinib, on the other hand, has approval for pediatric graft vs. host disease and provides downstream inhibition of TLR signaling [35]. Emavusertib is in clinical trials for lymphoma and leukemia and crosses the blood–brain barrier. If TLR4 pathway inhibitors can benefit MPS IIIB patients, they could make a profound difference in patient and family quality of life, not only directly from the drug but also from the benefit of encouraging newborn screening for MPS IIIB. Future studies are required to establish the value of these agents and if the TLR4 pathway plays a role in bone pathophysiology in the disease.

## 4. Materials and Methods

### 4.1. Sanfilippo Syndrome Type-B Mouse Colony

Our MPS IIIB mouse colony was graciously provided by Dr. Elizabeth Neufeld and Dr. Mark Sands and has been previously described and maintained at the University of Florida Biomedical Sciences vivarium under the oversight of the University of Florida (UF) Animal Care Services. The mice were fed a standard rodent diet and were given a 12 h biphasic light cycle. They were provided with unrestricted water access through an automatic dispenser.

### 4.2. Euthanasia and Tissue Collection

The mice were euthanized by CO_2_ inhalation followed by thoracotomy (N = 8–18 per group were males; N = 8–16 per group were females). Skin and fur over the thigh were removed, and each thigh muscle group was spread using dissection scissors as tissue spreaders. Both femurs were dissected and separated proximally from the acetabulum of the hips and distally, at the knee joints, from the tibiae. Femurs were then stripped of musculature, leaving the periosteum intact. Immediately after this procedure, femurs were fixed in 10% phosphate-buffered formalin at 4 °C for 48 h, placed in 70% ethanol, and stored at 4 °C until analysis. 

### 4.3. Quantitative Computed Tomography (pQCT)

The pQCT analysis was conducted on femurs prior to the biomechanical stress test. The femurs were thawed to room temperature and remained wrapped in saline-soaked gauze, except during scanning, using a Stratec XCT Research M instrument (Norland Medical Systems; Fort Atkinson, WI, USA), using the company software version 5.40. Scans were taken at a distance of 25% femur length proximal to the distal end of the femur (distal epiphysis) and mid-diaphysis. The former site is located at the secondary spongiosa of the distal femoral metaphysis. The mid-diaphyseal site is located at 50% of the femur’s average height corresponding to the diaphysis’s cortical bone. BMC, vBMD, and bone area were determined for total bone (trabecular and cortical bone) at the distal metaphysis and mid-diaphysis using methods previously described [36]. 

### 4.4. Biomechanical Strength Tests

The biomechanical properties of the femurs were conducted using an 858 Mini Bionix^®^ II (Model 359) hydraulic press. For this purpose, axial force challenged the femurs using a three-point bend method with a fixed span width of 10 mm. The axial force was applied with a 3 mm diameter cylindrical. Force was applied perpendicular to the femoral midshaft with a displacement rate of 0.05 mm/s until failure. Force and displacement were recorded temporally using TestStar II Software version 4.0 at a sampling rate of 205 Hz. Biomechanical properties from the 3-point bend were derived (Figure 9) using methods previously described (N/mm), ultimate load (N), work to fracture (Nmm), elastic modulus (Mpa), and ultimate stress (Mpa) [37].

### 4.5. Statistics

All pQCT, biomechanical, and histomorphometry variables were analyzed within each age group (three months, six months, and twelve months) using two-way ANOVA, considering independent variables, sex (male and female), and genotype (MPS IIIB and normal phenotype) using GraphPad Prism (version 10.0.2; Boston, MA, USA). For radiological data (pQCT), anomalies were excluded by removing samples with values two standard deviations from the mean for each respective experimental group [37]. All figures were created with BioRender.com under an academic license.

## 5. Conclusions

We present the first characterization of long-bone characteristics in the MPS IIIB C57BL/6 mouse model and suggest it may have utility as a model for the correction of osteological manifestations of MPS IIIB. Similar to MPS VII mouse models, MPS IIIB mice demonstrated increased vBMD in the femur, and this correlated to biomechanical stress characteristics. The use of these metrics for preclinical therapies may allow for the monitoring the amelioration of orthopedic complications in MPS IIIB.

## Figures and Tables

**Figure 1 ijms-24-13988-f001:**
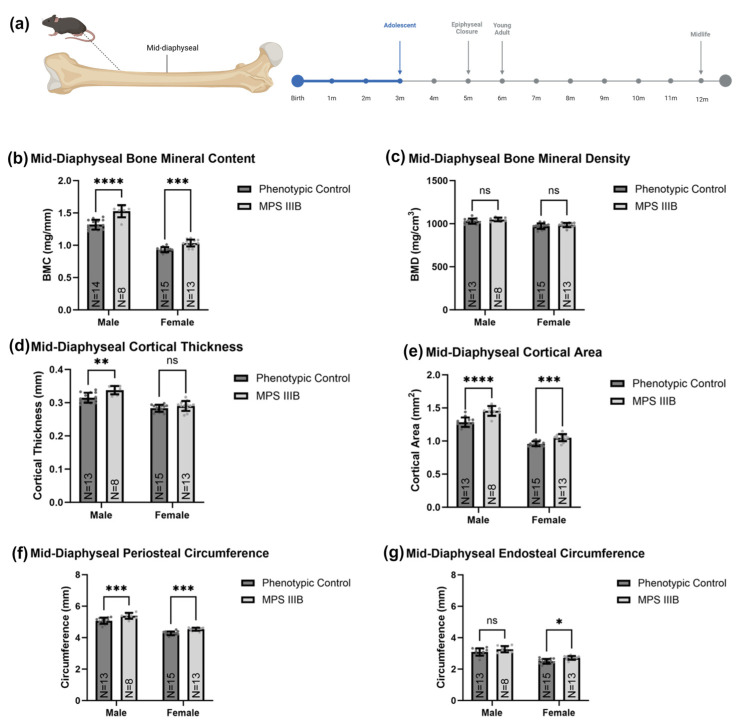
Metaphyseal pqCT metrics of MPS IIIB mice before epiphyseal closure. The data shown (**a**) are from ex vivo femur analysis of 3-month-old C57BL/6 mice with MPS IIIB (*NAGLU* −/−) or MPS IIIB asymptomatic carrier (*NAGLU* +/−). The graphs display mid-diaphyseal BMC (**b**), BMD (**c**), cortical thickness (**d**), cortical area (**e**), periosteal circumference (**f**), and endosteal circumference (**g**). Two-way ANOVA was performed on each radiological metric considering sex and genotype and the interaction between the two. Post hoc analysis was performed with a Dunn–Šidák correction. Error bars show standard deviation. ns = not significant; * *p* < 0.05; ** *p* < 0.01; *** *p* < 0.001; **** *p* < 0.0001.

**Figure 2 ijms-24-13988-f002:**
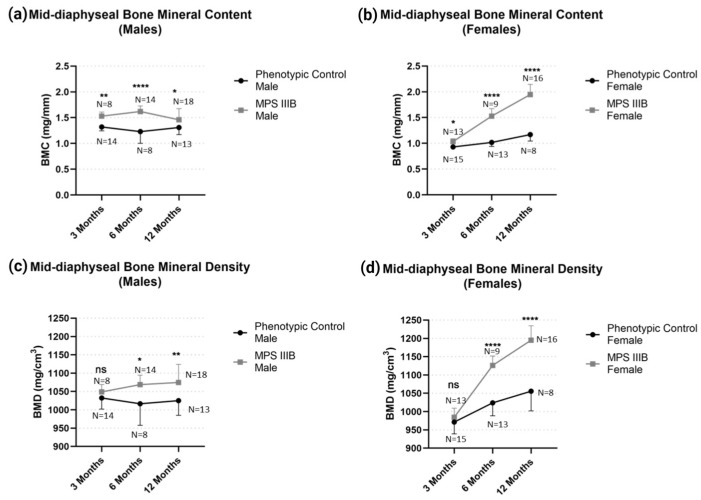
Temporal effect of diaphyseal mineralization in MPS IIIB mice from 3 to 12 months of age. Mid-diaphyseal BMC for (**a**) males and (**b**) females and BMD for (**c**) males and (**d**) females demonstrate significant differences. Two-way ANOVA was performed on each radiological metric. Post hoc analysis was performed with a Dunn–Šidák correction. Error bars show standard deviation. ns = not significant, * *p* < 0.05; ** *p* < 0.001; **** *p* < 0.0001.

**Figure 3 ijms-24-13988-f003:**
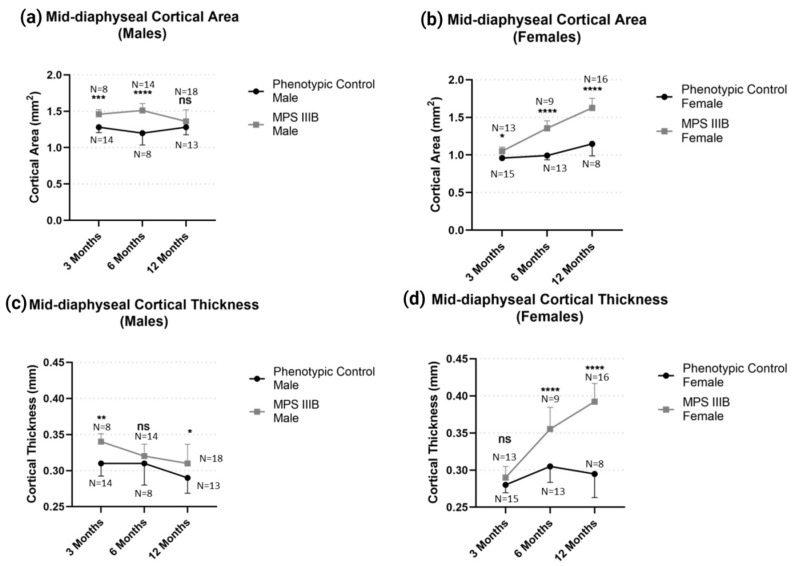
Temporal effect of diaphyseal gross morphology in MPS IIIB mice from 3 to 12 months of age. Mid-diaphyseal cortical area for (**a**) males and (**b**) females and cortical thickness for (**c**) males and (**d**) females demonstrate significant differences. Two-way ANOVA was performed on each radiological metric. Post hoc analysis was performed with a Dunn–Šidák correction. Error bars show standard deviation. ns = not significant, * *p* < 0.05; ** *p* < 0.01; *** *p* < 0.001; **** *p* < 0.0001.

**Figure 4 ijms-24-13988-f004:**
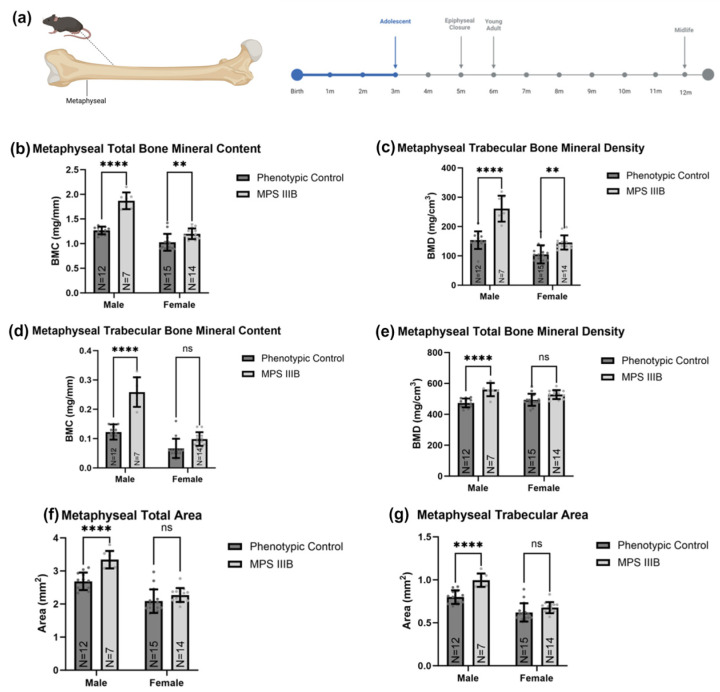
Metaphyseal pQCT metrics of MPS IIIB mice before epiphyseal closure. The data shown (**a**) are from ex vivo femur analysis of 3-month-old C57BL/6 mice with MPS IIIB (*NAGLU* −/−) or MPS IIIB asymptomatic carrier (*NAGLU* +/−). The graphs display metaphyseal total BMC (**b**), total BMD (**c**), trabecular BMC (**d**), trabecular BMD (**e**), total area (**f**), and trabecular area (**g**). Two-way ANOVA was performed on each radiological metric. Post hoc analysis was performed with a Dunn–Šidák correction. Error bars show standard deviation. ns = not significant; ** *p* < 0.01; **** *p* < 0.0001.

**Figure 5 ijms-24-13988-f005:**
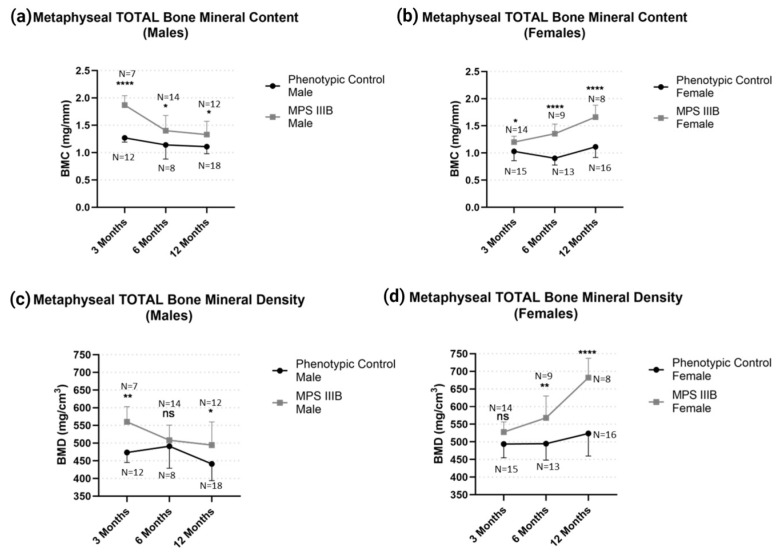
Temporal effect of metaphyseal total BMD and BMC in MPS IIIB mice from 3 to 12 months of age. Metaphyseal total BMC for (**a**) males and (**b**) females and metaphyseal total BMD for (**c**) males and (**d**) females demonstrate significant differences. Two-way ANOVA was performed on each radiological metric. Post hoc analysis was performed with a Dunn–Šidák correction. Error bars show standard deviation. ns = not significant; * *p* < 0.05; ** *p* < 0.01; **** *p* < 0.0001.

**Figure 6 ijms-24-13988-f006:**
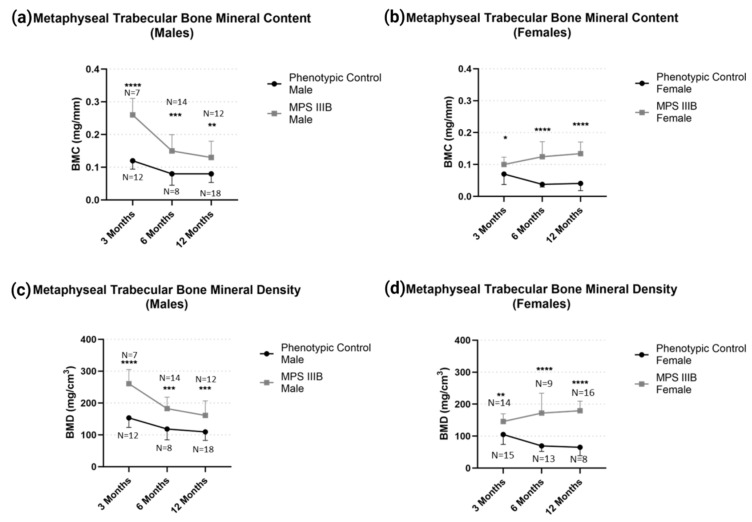
Temporal effect of metaphyseal trabecular BMD and BMC in MPS IIIB mice from 3 to 12 months of age. Metaphyseal trabecular BMC for (**a**) males and (**b**) females and BMD for (**c**) males and (**d**) females demonstrate significant differences. Two-way ANOVA was performed on each radiological metric. Post hoc analysis was performed with a Dunn–Šidák correction. Error bars show standard deviation. * *p* < 0.05; ** *p* < 0.01; *** *p* = 0.001; **** *p* < 0.0001.

**Figure 7 ijms-24-13988-f007:**
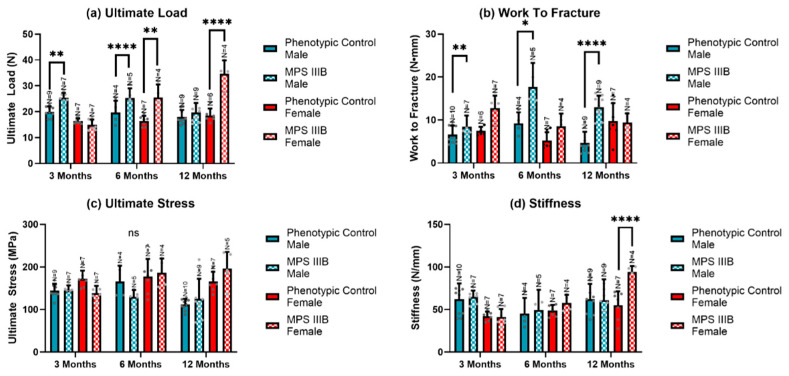
The biomechanical properties of an MPS IIIB mouse femur subjected to a 3-point bend axial stress test from 3 to 12 months of age. Ultimate load (**a**), work to fracture (**b**), ultimate stress (**c**), and stiffness (**d**) are shown (N = 4–10 per group). Statistically significant differences were noted for some groups, particularly ultimate stress for both sexes and work to fracture for males. A two-way ANOVA was performed on each biomechanical property. Post hoc analysis was performed with a Dunn–Šidák correction. Error bars show the standard deviation. * *p* < 0.05; ** *p* < 0.01; **** *p* < 0.0001, ns = not significant.

**Figure 8 ijms-24-13988-f008:**
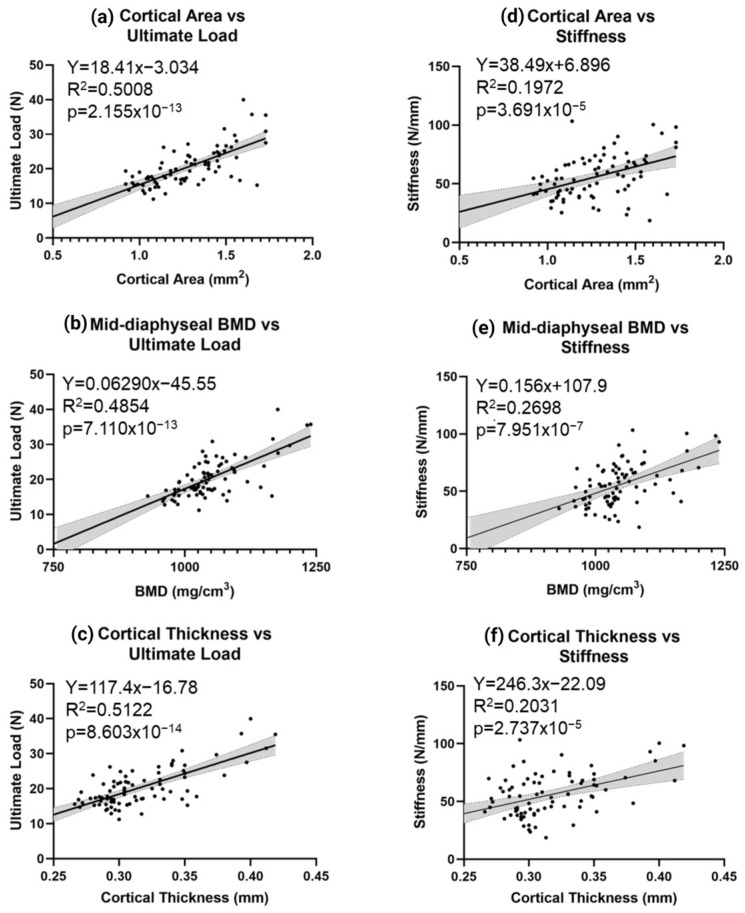
Correlation of radiological metrics to biomechanical properties. Correlation of ultimate load and stiffness to pQCT metrics was demonstrated for the cortical area (**a** and **d**, respectively), mid-diaphyseal BMD (**b** and **e**, respectively), and cortical thickness (**c** and **f**, respectively). Slope equations, R-squared, and *p*-values are included for each correlation. N = 80 for all graphs shown; 95% confidence intervals are shown in grey.

**Figure 9 ijms-24-13988-f009:**
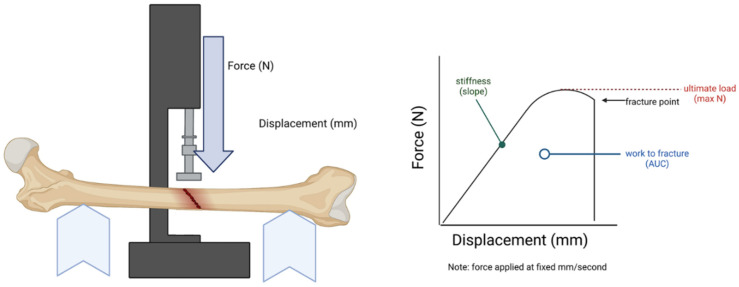
Biomechanical stress was measured using a three-point bend with metrics of stiffness, load, fracture point, and work to fracture measured as demonstrated above. Ultimate stress involves calculating the moment of inertia and is not demonstrated in the graph. N = Newtons; mm = millimeters; AUC = area under the curve.

## Data Availability

The data used for the project are available upon reasonable request.

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
