# Peer review of "Femoral Structure and Biomechanical Characteristics in Sanfilippo Syndrome Type-B Mice"

_ijms, 2023, doi:10.3390/ijms241813988_

Round 1

Reviewer 1 Report

Ashby and co-workers present an extensive, well-described characterization of the long-bone phenotype in MPS IIIB C57BL/6 mice.

This is an original report, addressing one of the issues, which tends to be less studied in MPS III: the skeletal phenotype. In fact, while most of MPS III distinctive phenotypes tend to be neurological, bone pathology has also been recognized in several patients. And if it is true that there is no treatment for the neurological phenotypes, the same applies to the skeletal ones. Most importantly, even for the diseases that do have a treatment (namely ERT), it fails to correct the bone phenotype.

This paper addresses a relevant issue, providing quantitative, well-analysed data, which may ultimately be of use not only for other researchers (bone-related clinical endpoints for therapeutic efficacy assessments, p.e.) but also for clinicians in general (as it may reflect the human phenotype). While a few technical criticisms may be presented (e.g.: physical activity was not analysed), the authors comment on those flaws themselves, and still provide a well-grounded, nice discussion section.

Still, there are a few minor issues I would like to see addressed:

On the Introduction section

-          page 1, line 34: the authors state that MPS diseases are classified based on which GAG pathway is affected. That is true, obviously, but as it is, their statement may be misleading as there is no reference to the biochemical defects underlying them. I mean: different MPS accumulate the same GAGs, and in order to classify them properly, the enzymatic defect must be unveiled. I would recommend the authors to include a few more information on this section.

-          Page 2, line 45: the authors provide a nice description on the role HS plays in TLR-4 stimulation, but make no reference to the type of studies where that association was found: where was it seen? In patients? In mice? A few extra details would be nice.

-          Page 2, line 51: the authors comment on ERT/gene therapy over neurological disease, but make no reference on its effects over bone disease. Given the topic of this manuscript, I believe it would be nice if they refer to the bone phenotype here. Is there any data? If so, it should be referred to; if not, is should still be commented on.

-          Page 2, line 66: instead of closing the introduction with a highlight on their working hypothesis alone, I would strongly recommend the authors to add a brief statement on their general conclusions after this study.

On the Discussion section

-          Page 8, lines 215-116: just out of curiosity but do the authors have any clue on why male femurs take longer to break the all the other groups?

Minor issues:

-          Abstract, line 15: please include the acronym ERT, as it is further referred in this section

-          Introduction, page 2, line 44: please use the acronym GAG instead of the full name

-          Introduction, page 2, line 48/49: please use the acronym ERT instead of the full name

-          Throughout the manuscript: there is always something wrong with the way references are placed in the text. Instead of showing up before the ‘.’, they always come right after it. Please correct.

No major issues, really. Overall, this is a well written, easy to follow manuscipt.

Author Response

Reviewer 1

This is an original report, addressing one of the issues, which tends to be less studied in MPS III: the skeletal phenotype. In fact, while most of MPS III distinctive phenotypes tend to be neurological, bone pathology has also been recognized in several patients. And if it is true that there is no treatment for the neurological phenotypes, the same applies to the skeletal ones. Most importantly, even for the diseases that do have a treatment (namely ERT), it fails to correct the bone phenotype.

This paper addresses a relevant issue, providing quantitative, well-analysed data, which may ultimately be of use not only for other researchers (bone-related clinical endpoints for therapeutic efficacy assessments, p.e.) but also for clinicians in general (as it may reflect the human phenotype). While a few technical criticisms may be presented (e.g.: physical activity was not analysed), the authors comment on those flaws themselves, and still provide a well-grounded, nice discussion section.

We thank you for the thoughtful and well-read review. Your time, expertise and input have been greatly appreciated in making this manuscript better than where it began.

page 1, line 34: the authors state that MPS diseases are classified based on which GAG pathway is affected. That is true, obviously, but as it is, their statement may be misleading as there is no reference to the biochemical defects underlying them. I mean: different MPS accumulate the same GAGs, and in order to classify them properly, the enzymatic defect must be unveiled. I would recommend the authors to include a few more information on this section.

I have clarified this overlooked ambiguity and expounded upon this section of the background. Thank you for catching this oversight.

-          Page 2, line 45: the authors provide a nice description on the role HS plays in TLR-4 stimulation, but make no reference to the type of studies where that association was found: where was it seen? In patients? In mice? A few extra details would be nice.

It was my pleasure to clarify the details of the older studies and cite some of the work that established this connection, and lead to some of my hypotheses for future work in the discussion. The initial introduction draft was too lean on this topic.

-          Page 2, line 51: the authors comment on ERT/gene therapy over neurological disease, but make no reference on its effects over bone disease. Given the topic of this manuscript, I believe it would be nice if they refer to the bone phenotype here. Is there any data? If so, it should be referred to; if not, is should still be commented on.

I have expounded upon this and included much of the limited data on bone manifestations in MPS III, which is heavily reliant on case series.

-          Page 2, line 66: instead of closing the introduction with a highlight on their working hypothesis alone, I would strongly recommend the authors to add a brief statement on their general conclusions after this study.

I have expanded the conclusion section (section 5) to provide more on this.

On the Discussion section

-          Page 8, lines 215-116: just out of curiosity but do the authors have any clue on why male femurs take longer to break the all the other groups?

I included my speculations and hypotheses for these male and female disparities in the discussion section.

Minor issues:

-          Abstract, line 15: please include the acronym ERT, as it is further referred in this section

This has been fixed, thank you for noting this.

-          Introduction, page 2, line 44: please use the acronym GAG instead of the full name

This has been fixed.

-          Introduction, page 2, line 48/49: please use the acronym ERT instead of the full name

This has been fixed.

-          Throughout the manuscript: there is always something wrong with the way references are placed in the text. Instead of showing up before the ‘.’, they always come right after it. Please correct.

I have updated the citations to follow this format.

Reviewer 2 Report

Frederick et al.'s research highlights notable radiological and biomechanical differences in the femurs of MPS IIIB C57BL/6 mice compared to their control counterparts. This study underscores the mouse model's utility in elucidating the osteological implications of MPS IIIB and emphasizes the need for further exploration of TLR4 pathway inhibitors as potential therapeutic measures. While the paper is thought-provoking, some aspects warrant revision before it can be accepted. My recommendations are as follows:

  1. How do bone manifestations in MPS IIIB compare to other MPS types, particularly concerning femoral structure? A comparative analysis might yield richer insights.
  2. For Figures 1, 4, and 7, it would enhance clarity if the authors displayed data points representing each individual mouse, providing readers with a more detailed view.
  3. Some sentences in the manuscript are lengthy and could be simplified for reader accessibility. For instance: Original: "The results ultimately demonstrated detectable physical changes in the femurs of MPS IIIB mice compared to age and sex-matched controls, with variable phenotype depending on the sex of the mouse and the age of the mouse." Revised: "The results revealed discernible physical changes in the femurs of MPS IIIB mice. These changes were influenced by both the mouse's sex and age, especially when juxtaposed with age and sex-matched controls."
  4. Expanding on the ramifications of the study's conclusions could be beneficial. Specifically, what is the significance of the increased femoral BMD observed in male MPS IIIB mice? How might these insights influence potential treatment avenues or inform our grasp of the disease's trajectory in human patients?
  5. The paper indicates gender-based disparities in the femoral attributes of MPS IIIB mice. A deeper discussion into the underlying reasons for these differences would be valuable.
  6. Have the authors investigated the serum bone metabolism biomarkers for the MPS IIIB mice and controls? If such data exists, it would be an insightful addition to the paper.

The manuscript is generally well-written and the language used is academic and appropriate for a scientific research paper. However, there are a few areas where the texts are quite long, which may make them harder for readers to follow. Consider breaking some of them down for clarity.

Author Response

Reviewer 2

Frederick et al.'s research highlights notable radiological and biomechanical differences in the femurs of MPS IIIB C57BL/6 mice compared to their control counterparts. This study underscores the mouse model's utility in elucidating the osteological implications of MPS IIIB and emphasizes the need for further exploration of TLR4 pathway inhibitors as potential therapeutic measures. While the paper is thought-provoking, some aspects warrant revision before it can be accepted. My recommendations are as follows:

1. How do bone manifestations in MPS IIIB compare to other MPS types, particularly concerning femoral structure? A comparative analysis might yield richer insights.

I have included more citations in the introduction about how MPS III compares to other MPS types, as MPS III is generally regarded as having milder symptoms and therefore gets less attention. However, these symptoms are still debilitating for patients and caregivers, which hopefully is a more clear point in the introduction now.

2. For Figures 1, 4, and 7, it would enhance clarity if the authors displayed data points representing each individual mouse, providing readers with a more detailed view.

All of the figures have been updated as requested.

3. Some sentences in the manuscript are lengthy and could be simplified for reader accessibility. For instance: Original: "The results ultimately demonstrated detectable physical changes in the femurs of MPS IIIB mice compared to age and sex-matched controls, with variable phenotype depending on the sex of the mouse and the age of the mouse." Revised: "The results revealed discernible physical changes in the femurs of MPS IIIB mice. These changes were influenced by both the mouse's sex and age, especially when juxtaposed with age and sex-matched controls."

I have updated the manuscript to reflect these changes and enhance readability.

4. Expanding on the ramifications of the study's conclusions could be beneficial. Specifically, what is the significance of the increased femoral BMD observed in male MPS IIIB mice? How might these insights influence potential treatment avenues or inform our grasp of the disease's trajectory in human patients?

I have expanded the conclusion to make it more clear specifically what the implications are for human patients, as this is mostly relevant for future preclinical trials. Since these MPS IIIB mouse models are quite well-utilized in the research community, it means these bone metrics can be monitored for any future therapies tested.

5. The paper indicates gender-based disparities in the femoral attributes of MPS IIIB mice. A deeper discussion into the underlying reasons for these differences would be valuable.

I have expanded my discussion, speculation, and some preliminary hypotheses to explain these discrepancies, outside of the possibility of physical activity as a confounder. This was similar to Reviewer 1’s comment, and I made certain to provide some grounds for my discussion on the topic.

6. Have the authors investigated the serum bone metabolism biomarkers for the MPS IIIB mice and controls? If such data exists, it would be an insightful addition to the paper.

Yes, we certainly have considered this, and we recently got a grant to investigate TLR-4 targeting interventions which will include serum biomarkers (such as IL-1, TNF-alpha, etc). Unfortunately for this initial study, we were not able to collect serum for biomarkers due to limited resources, similar to the reason why we were not able to control for physical activity. We have no serum data on the mice from the study in this manuscript, but hopefully once the next study is complete we will be able to explore this.